# Spatial Distribution and Biochemical Characterization of Serine Peptidase Inhibitors in the Venom of the Brazilian Sea Anemone *Anthopleura cascaia* Using Mass Spectrometry Imaging

**DOI:** 10.3390/md21090481

**Published:** 2023-08-30

**Authors:** Daiane Laise da Silva, Rodrigo Valladão, Emidio Beraldo-Neto, Guilherme Rabelo Coelho, Oscar Bento da Silva Neto, Hugo Vigerelli, Adriana Rios Lopes, Brett R. Hamilton, Eivind A. B. Undheim, Juliana Mozer Sciani, Daniel Carvalho Pimenta

**Affiliations:** 1Programa de Pós-Graduação em Ciências-Toxinologia, Instituto Butantan, Av. Vital Brasil 1500, Butantã, São Paulo 05503-900, Brazil; emidio.beraldo@butantan.gov.br (E.B.-N.); guilherme.coelho@butantan.gov.br (G.R.C.); hugo.barros@butantan.gov.br (H.V.); adriana.lopes@butantan.gov.br (A.R.L.); 2Laboratório de Bioquímica, Instituto Butantan, Av. Vital Brasil 1500, São Paulo 05503-900, Brazil; rodrigovalladao3@gmail.com (R.V.); oscar.bentoneto@gmail.com (O.B.d.S.N.); 3Centre for Advanced Imaging, University of Queensland, St. Lucia, QLD 4072, Australia; e.a.b.undheim@ibv.uio.no; 4Laboratório de Genética, Instituto Butantan, Av. Vital Brasil 1500, São Paulo 05503-900, Brazil; 5Centre for Microscopy and Microanalysis, University of Queensland, St. Lucia, QLD 4072, Australia; b.hamilton@uq.edu.au; 6Centre for Ecological and Evolutionary Synthesis, Department of Biosciences, University of Oslo, 0316 Oslo, Norway; 7Laboratório de Farmacologia Molecular e Compostos Bioativos, Universidade São Francisco, Av. São Francisco de Assis, 218, São Paulo 12916-900, Brazil; juliana.sciani@usf.edu.br

**Keywords:** serine peptidase inhibitors, venom, sea anemones, mass spectrometry imaging

## Abstract

Sea anemones are known to produce a diverse array of toxins with different cysteine-rich peptide scaffolds in their venoms. The serine peptidase inhibitors, specifically Kunitz inhibitors, are an important toxin family that is believed to function as defensive peptides, as well as prevent proteolysis of other secreted anemone toxins. In this study, we isolated three serine peptidase inhibitors named *Anthopleura cascaia* peptide inhibitors I, II, and III (ACPI-I, ACPI-II, and ACPI-III) from the venom of the endemic Brazilian sea anemone *A. cascaia*. The venom was fractionated using RP-HPLC, and the inhibitory activity of these fractions against trypsin was determined and found to range from 59% to 93%. The spatial distribution of the anemone peptides throughout *A. cascaia* was observed using mass spectrometry imaging. The inhibitory peptides were found to be present in the tentacles, pedal disc, and mesenterial filaments. We suggest that the three inhibitors observed during this study belong to the venom Kunitz toxin family on the basis of their similarity to PI-actitoxin-aeq3a-like and the identification of amino acid residues that correspond to a serine peptidase binding site. Our findings expand our understanding of the diversity of toxins present in sea anemone venom and shed light on their potential role in protecting other venom components from proteolysis.

## 1. Introduction

Animals belonging to the phylum Cnidaria all share one main characteristic: the presence of cnidocytes—specialized stinging cells—that produce and harbor a dischargeable venomous capsule called the nematocyst [1,2]. This phylum comprises more than 13,000 species, and approximately 10% correspond to sea anemones (Anthozoa: Actiniaria) [3,4,5]. Sea anemones are benthic, sessile cnidarians with a remarkably basic body plan that is covered by nematocysts [1,6,7]. These small structures release venom under mechanical or chemical stimuli and can be found in several parts of the sea anemone’s body, including its tentacles, actinopharynx, column, and mesenterial filaments [1,4,8].

Sea anemones exhibit an interesting structural diversity of toxins, spanning multiple cysteine-rich peptide scaffolds stabilized by multiple intra-molecular disulfide bridges [6,9]. Such toxins can affect at least 20 types of pharmacological targets, including several subtypes of voltage-gated sodium (Nav) and potassium (Kv) channels, Acid-sensing channels, and transient receptor potential (TRP) channels, which highlights their potential for drug discovery [4,9]. However, this pharmacological potential is also a reflection of the diverse roles that these toxins play in sea anemones’ day-to-day life, which include predation, defense, prey immobilization, digestion, and intra-specific competition. While some of these toxins, such as those targeting Nav and Kv channels, are known to immobilize prey by inhibiting voltage-gated ion channels [7,10], other toxins, like actinoporins and Membrane Attack Complex/Perforin (MACPF)-type cytolysins, create pores in cell membranes, causing necrosis in competitors [11,12,13]. Additionally, some of these toxins might impair the activity of target enzymes, inhibiting the peptidases that otherwise would impair the action of venom’s components, acting as defense toxins [7,14].

Serine peptidase inhibitors have been found in sea anemones, and it has been suggested that they are ultimately employed to protect other venom components from degradation caused by endogenous or exogenous enzymes [7]. Interestingly, several of these inhibitors have been classified as both Kunitz-type inhibitors and type 2 potassium channel toxins due to their dual ability to both inhibit serine peptidases and block Kv channels [4,9,15]. AsKC11 (*Anemonia sulcate*), HCRG1 and HCRG2 (*Heteractis crispa*), and ShPI-1 (*Stichodactyla helianthus*) are examples of these Kunitz peptide toxins [16,17,18]. The bifunctional activity of such peptides suggests that they are possibly involved in defense when deactivating exogenous enzymes and aggression when paralyzing prey [7,19].

*Anthopleura* is among the most familiar genera of sea anemones and is frequently found between rocky intertidal communities in both temperate and tropical zones worldwide [20]. This genus comprises more than 50 reviewed species according to the World Register of Marine Species—WoRMS (as of 25 May 2022) [21], and exhibits two common morphological aspects: The presence of verrucae (adhesive structures) throughout the entire column, and acrorhagi—dense bulbous structures filled with nematocysts—commonly used for intraspecific competition [20,22]. Interestingly, these species are morphologically differentiated both by the abundance and distribution of verrucae on columns and by the distribution and types of nematocysts [23,24].

In the Southwest Atlantic region, *Anthopleura* species are found in the shallow waters of Brazilian and Caribbean shorelines. Brazil shelters three of the five species found: *A. krebsi*, *A. varioarmata*, and *A. cascaia* [25]. Of these, *A. cascaia* is endemic to Brazil, where it is often found between rocky shores, more specifically in mid and inferior eulittoral and infralittoral zones: over and under rocks, sand, tide channels, tide pools, caves, and clefts [21]. During low tide periods, this species is easily spotted by the presence of gravel and small rocks attached to its adhesive verrucae (Figure 1).

To date, studies involving *A. cascaia* have typically been related to trophic ecology and biogeographical distribution [25,26,27], with only a single study reporting the characterization of neurotoxic fractions from this species [28]. This biochemical study revealed that the venom of this sea anemone possesses three peptides that selectively inhibit different Nav and Kv channels expressed on *Xenopus* oocytes: AcaIII1425 (3337 Da), AcaIII2970 (4881 Da) and AcaIII3090 (4880 Da) [28]. Thus, despite being such a common species, the venom of *A. cascaia* remains underexplored [4].

Here, we report that the venom of the Brazilian sea anemone *A. cascaia* is a rich source of serine peptidase inhibitors, ranging between 2.6 kDa and 4.7 kDa, with inhibitory activity against trypsin. Furthermore, we show the spatial distribution of these toxins across *A. cascaia* using mass spectrometry imaging (MSI), revealing their presence mainly in the tentacles, the pedal disc, and the mesenterial filaments from the sea anemone.

## 2. Results

In this work, we describe the presence of serine peptidase inhibitors in the venom of *A. cascaia* (Figure 1) and their isolation using reversed-phase liquid chromatography. To characterize these peptides, a protocol comprising four steps was followed: 1. fractionation of the venom by RP-HPLC; 2. enzymatic inhibition determination; 3. mass spectrometry biochemical characterization by MALDI-TOF, LC-MS, and LC-MS/MS; and 4. mass spectrometry imaging (MSI). For this study, trypsin was used as the model serine peptidase for tracking potential serine peptidase inhibitors. Crude *A. cascaia* venom demonstrated 100% inhibition of trypsin. The crude venom was analyzed using SDS-PAGE and MALDI-TOF/MS to provide an initial molecular characterization (Appendix A).

Subsequently, the venom of *A. cascaia* was fractionated using RP-HPLC to track serine peptidase inhibitors. As shown in Figure 2A, the chromatographic profile of the venom was divided into six fractions (F1 to F6), and the majority of venom components eluted between 30% and 60% B (15 to 22 min), mainly corresponding to F3. The fractions were assayed for trypsin inhibition and analyzed by SDS-PAGE and MALDI-TOF (Figure 2B,C). Only F3 and F4 exhibited the presence of serine peptidase inhibitors. F3 showed complete inhibition of trypsin, and SDS-PAGE revealed that the majority of the components were 14 and 20 kDa (Figure 2B). MALDI-TOF analysis of F3 revealed the presence of ions with *m*/*z* 4864, 4902, 5095, and 6663 corresponding to the most abundant ions (Figure 2C). As F3 exhibited the greatest response in the trypsin inhibition assay, F3 was chosen for further purification of potential serine peptidase inhibitors (Appendix A).

F3 was further fractionated into nine subfractions (F3.1 to F3.9) using reversed-phase chromatography (Figure 3A). Subfractions F3.7 and F3.8 exhibited complete trypsin inhibition (100%) (Figure 3 and Appendix A), and their mass spectra are shown in Figure 3B. To isolate the active inhibitors, F3.7 and F3.8 were further fractionated using a gradient of 5–60% B in 40 min, and the subfractions were tested for their trypsin inhibitory activity. All subfractions showed some degree of trypsin inhibition, with the highest inhibition levels (>90%) observed in F3.7.5 and 3.7.6 (Figure 4B). The table containing the remaining trypsin activity for all subfractions is presented in Appendix A. MALDI-TOF analysis revealed that F3.7.6 contained a single peptide with *m*/*z* of 4461, named ACPI-I (*A. cascaia* peptide inhibitor-I), while F3.7.5 contained three main components with similar *m*/*z*: 4518, 4685, and 4755 (Figure 4C). In addition, subfractions from F3.8, specifically F3.8.4, F3.8.6, and F3.8.8, which showed the most isolated components, presented 85%, 65%, and 59% inhibition of trypsin, respectively (Figure 5B). MALDI-TOF analysis revealed that F3.8.4 contained three components observed at *m*/*z* 2617, 2633, and 2649. The *m*/*z* 16 spacing observed may suggest that the peptide is rich in easily oxidized residues such as methionine. This peptide was named ACPI-II (*A. cascaia* peptide inhibitor-II). F3.8.8 was also analyzed by MALDI-TOF, revealing an *m*/*z* of 2649 and was named ACPI-III (*A. cascaia* peptide inhibitor-III). All fractions were tested for trypsin inhibition, and the results are summarized in Appendix A.

LC-MS analysis provided a comprehensive mass profile of the serine peptidase inhibitors isolated from the venom fractions. Figure 6 depicts the mass spectra of peptides obtained by LC-MS analysis for each inhibitory fraction. Notably, F3.7.6 presented a distinct peptide with a mass of 4465.70 ± 0.16 Da corresponding to ACPI-I, while F3.8.4 exhibited 2652.45 Da and 2668.60 Da peptides likely corresponding to the same peptide (ACPI-II) with a mass difference of 16 *m*/*z*; F3.8.8 displayed a peptide with a mass of 2641.97 Da (ACPI-III). F3.7.5 is the sole fraction that appears to contain two primary peptide inhibitors (4645.84 ± 0.16 Da and 4482.36 ± 0.16 Da).

To identify the number of cysteines present in the isolated peptides (ACPI-I, ACPI-II, and ACPI-III), these molecules were reduced with Dithiothreitol (DTT) and then alkylated with Iodoacetamide, which resulted in an increase in mass of 57 Da per cysteine present in the peptide. The resulting changes in mass were analyzed by MALDI-TOF. ACP-I exhibited a mass increase of *m*/*z* 342, indicating the presence of six cysteines, while ACP-II and ACPI-III exhibited mass increases of *m*/*z* 228 and *m*/*z* 285, respectively, indicating the presence of four and five cysteines in each inhibitor (Figure 7). This result indicates that it is likely that disulfide bonding plays a part in the structure of these peptides, which will undoubtedly have a role in the function of these peptides in the venom of *A. cascaia*.

The isolated peptides ACPI-I, ACPI-II, and ACPI-III were further analyzed using LC-MS/MS, which revealed that each showed some level of similarity to PI-Actitoxin-Aeq3a-like from *A. tenebrosa* (Table 1). The mass spectra obtained from this analysis are presented in Table 1, where a Blastp hit was identified in a search within the Cnidaria database, and the sequence coverage for each peptide inhibitor was determined to be 37%, 46%, and 14% for ACPI-I, ACPI-II, and ACPI-III, respectively. The table also shows the supporting peptides identified, including two peptides found in ACPI-I, three in ACPI-II, and only one supporting peptide found in ACPI-III (Table 1; Appendix A). However, the construction of the sequence of these inhibitors based solely on De novo peptides (Figure 8) revealed that the peptides found for ACPI-II share 78% identity with PI-Actitoxin-Aeq3a-like, the peptides found in ACPI-III share 66% identity with this toxin, and ACPI-I presents a unique de novo peptide (YYYDESLGHMER) that shares 87% identity with the YYLDETLGRLER peptide from PI-Actitoxin-Aeq3a-like (Appendix A).

To further investigate the ecological roles of the serine peptidase inhibitors, which are known to be distributed across functionally distinct tissues and regions in sea anemones, we conducted MALDI mass spectrometry imaging to map their distribution throughout the body of *A. cascaia*. The results, presented in Figure 9, revealed that the inhibitors isolated from F3.7.5 were mainly localized in the tentacles (*m*/*z* 4755 and *m*/*z* 4685) and pedal disc region (*m*/*z* 4518), suggesting that these peptides are likely involved in predation at the tentacles and defense in the pedal disc, considered a region related with substrate adhesion. The serine peptidase inhibitor found in F3.7.6 (*m*/*z* 4461) showed a moderate distribution in the pedal disc and mesenteric tissue, suggesting potential functions in adhesion and digestion, respectively. Peptide isolated from F3.8.4 (*m*/*z* 2633), which exhibited three possible oxidized *m*/*z*, was detected in the mesenteries and tentacles of the sea anemone, sharing a similar distribution with the peptide found in F3.8.8 (*m*/*z* 2642). These results suggest that both peptides are likely involved in prey capture and digestion.

## 3. Discussion

In the current study, three peptides (ACPI-I, ACPI-II, and ACPI-III) from the venom of the sea anemone *A. cascaia* were isolated, and an enriched fraction (F3.7.5) capable of binding trypsin and impairing its activity was identified. Such peptides, classified here as serine peptidase inhibitors, showed from 59% to up to 93% inhibition of trypsin’s activity. To achieve purification, three reversed-phase chromatographic separations were required.

Sea anemones have been shown to contain serine peptidase inhibitors, with the venom Kunitz-type family being one of the four main toxin families found in these animals [4]. Kunitz toxins have been described as important components of cnidaria venoms, being used as defensive tools against aggressors or contributing to animal feeding through prey paralysis when acting as bifunctional molecules. Some Kunitz toxins have been shown as capable of inhibiting several serine peptidase representants at the same time that they can efficiently block voltage-gated potassium channels, especially Kv1.1, Kv1.2, and Kv1.6 [7,15,17,18].

The presence of such peptides in venom is typically identified through proteomics or transcriptomics analysis of secreted molecules or venom-related structures [29,30]. However, other strategies for identifying such inhibitors may involve individual characterization of components previously isolated using chromatographic techniques from venom or extracts of related structures [31,32,33]. This characterization typically involves testing the inhibitor’s direct effect on serine peptidase enzymes such as trypsin, chymotrypsin, and kallikreins, as well as the determination of their amino acid sequence [34,35]. Examples of Kunitz-type serine protease inhibitors isolated from sea anemones include PI-stichotoxin-Hmg3a (UniProt entry-C0HK72) from *Heteractis magnifica*, PI-stichotoxin-Hcr2g (UniProt entry-C0HJU7) from *H. crispa*, and ATPI-I (UniProt entry-A0A6P8HC43) and ATPI-II from *A. tenebrosa* [17,34,35].

While the venom of *A. cascaia* has been studied to some extent, with the potassium channel inhibitor Acatoxin 1 (UniProt entry A0A6I8WFP9/PDB entry-6NK9), and three neurotoxins (AcaIII1425, AcaIII2970, and AcaIII3090) having been identified [28,36,37], much remains unknown about this species’ venom. However, at the genus level, over 30 toxins have been annotated for *Anthopleura* in Swiss-Prot/UniProtKB, including Delta-actitoxin-Axm1a (Anthopleurin A) and Delta-actitoxin-Axm1b (Anthopleurin B); search performed 2 May 2022 [38,39,40,41].

Within this genus, six Kunitz-type protease inhibitors have been characterized in three species. The inhibitors PI-actitoxin-Axm2a (P81547), PI-actitoxin-Axm2b (P81548), and PI-actitoxin-Axm2b (P0DMX0) were isolated from the aqueous extract of *A.* aff. *xanthogrammica*, and their average masses ranged from 6341.13 Da to 6979.68 Da [32,33]. Additionally, PI-actitoxin-Afv2a (6096.98 Da, UniProt entry-P0DMJ3) and PI-actitoxin-Afv2b (5994.93 Da, UniProt entry-P0DMJ4) were isolated from the acrorhagi of *A. fuscoviridis*; both of which showed strong antitryptic activity and contained a basic residue (Arg or Lys) at position 17 in their sequences [42]. Finally, the most recent Kunitz inhibitor described for this genus is KappaPI-actitoxin-Ael3a (UniProt entry-P86862), which was isolated from *A. elegantissima* and had a molecular mass of 7484.5 Da. This inhibitor exhibits dual activity, targeting both voltage-gated potassium channels (Kvs) and serine peptidases [43].

All the inhibitors isolated from *Anthopleura* species are classified as Kunitz type, exhibiting three disulfide bridges and mature chains of 56 to 61 residues. The serine peptidase inhibitors evaluated for *A. cascaia* in this study showed *m*/*z* values between 2633 and 4755 by MALDI-TOF and ESI analysis. These peptides exhibit molecular masses below the expected for Kunitz inhibitors found for this genus, which typically presents average masses around 6000 Da [32,33,42,43]. However, inhibitors with similar molecular masses have been described for the sea anemone *A. equina* and the snake *Daboia siamensis*. *A. equina* has two toxins belonging to the venom Kunitz-type family and Sea anemone type 2 potassium channel toxin subfamily, with molecular masses of 4071 Da (UniProt entry-P0DMW8) and 4308 Da (UniProt entry-P0DMW9) [31]. Similarly, *D. siamensis* venom contains two Kunitz inhibitors with molecular masses of 2050 Da (UniProt entry-P85040) and 2691 Da (UniProt entry-P85039) [39]. These findings suggest that Kunitz-type inhibitors with similar molecular masses can be found in distantly related organisms.

Transcriptome and proteome analyses have revealed the presence of serine peptidase inhibitors in *A. elegantissima* and *A. dowii.* In *A. elegantissima*, the transcriptome analysis of aggressive and non-aggressive polyps has shown that venom Kunitz/Kv2 toxin transcripts are present in both types of polyps, but such toxins are highly expressed in the acrorhagi of aggressive polyps [44]. Additionally, twelve transcripts belonging to the venom Kunitz/Kv2 toxin family have been identified in the tentacle’s transcriptome of *A. dowii*. Five of these transcripts have shown similarity to the mature KappaPI-AITX-Ael3a from *A. elegantissima*, exhibiting 37% to 80% identity with this toxin. Another five transcripts have demonstrated similarity to serine peptidase inhibitors described for snake species such as *Walterinnesia aegyptia*, *D. russelii*, and *Vipera ammodytes ammodytes*. At the proteome level, only the transcript c14874_g1 presenting similarity to KappaPI-actitoxin-Ael3a (UniProt entry-P86862) from *A. elegantissima* was identified in the mucus [8]. Apart from these Kunitz-type inhibitors, two Kazal-type transcripts have been identified in the transcriptome of *A. dowii*, exhibiting 41% identity with a turripeptide LoI9.1 (UniProt entry-P0DKM7) from the sea snail *Lophiotoma olangoensis* and 75% identity with PI-actitoxin-Avd5a from *Anemonia sulcata* (UniProt entry-P16895) [8].

The inhibitors isolated from *A. cascaia* venom, namely ACPI-I, ACPI-II, and ACPI-III, showed similarity to PI-Actitoxin-Aeq3a-like from *A. tenebrosa* when analyzed by LC-MS/MS. This Aeq3a-like toxin, predicted by computational analysis, contains a putative serine protease binding site formed by the peptide ^11^TGRCMGYFP^19^ in addition to Glu^25^, Ile^34^, and Cys^38^ (numbering according to the processed peptide, as of 15 March 2023 via NCBI database/BLASTp search). The de novo peptide sequences identified in ACPI-I, ACPI-II, and ACPI-III were classified under the Pancreatic trypsin inhibitor Kunitz domain (IPR002223) when submitted to the InterPro Classification of Protein families database, exhibiting slight differences in the amino acid sequence of the Kunitz domain (as of 15 March 2023).

Glu^25^, Ile^34^, and Cys^38^ present in ACPI-I are also present in the putative serine peptidase binding domain of the predicted mature toxin Aeq3a-like (XP_031550082.1) (Figure 8 and Appendix A). Moreover, the analysis of the Cys content of ACPI-I peptide by MALDI-TOF reveals the presence of the six classic cysteine residues that form the Kunitz domain signature, as shown by the +57 Da shift per cysteine in the original peptide mass (Figure 7A). In contrast, the analysis of the ACPI-II sequence (Figure 8) indicates the presence of DLD residues instead of TGRC at positions 12, 13, and 14 of the serine peptidase binding domain, as well as a Lys at position 34 instead of Ile, as well as a Cys residue at position 38. For ACPI-III, the Glu is also present at position 25, but a Tyr residue replaces Ile at position 34 (Figure 8). MALDI-TOF analysis shows that these serine peptidase inhibitors (ACPI-II and ACPI-III) do not conform to the classical Kunitz peptide structure, as they contain only four and five cysteines each, and are stabilized for only two disulfide bridges (Figure 7). This observation may explain why ACPI-II and ACPI-III exhibit reduced capacity for binding trypsin, inhibiting only 65% and 59%, respectively, compared to ACPI-I, which shows 93% inhibition of the peptidase. Due to the genus relationship, sequence elucidation by LC-MS/MS analysis, and the proven existence of Kunitz inhibitors in the venom of related sea anemones, we suggest that the three isolated serine peptidase inhibitors *m*/*z* 4461 (ACPI-I), *m*/*z* 2633 (ACPI-II), and *m*/*z* 2642 (ACPI-III) belong to the Kunitz family.

Regarding the spatial distribution of the serine peptidase inhibitor toxins found in *A. cascaia*, this study provides novel insights through MSI. While previous studies have identified Kunitz toxins in crude extracts from sea anemone tentacles, our findings represent the first direct evidence of these inhibitors in their expected tissue–tentacles. Such structures are important tools used for sea anemones, as they are used for capturing prey and repelling predators, making the localization of these toxins particularly relevant to understanding their defensive and aggressive strategies [22,45]. Additionally, MSI analysis also revealed the unexpected presence of such inhibitors in the pedal disc of *A. cascaia*, a tissue usually related to the surface attachment of sea anemones [46]. Previous studies have shown that pedal tissue might present spirocysts and have even hypothesized the participation of these nematocysts in the adhesion process [47]; other studies have shown that spirocysts are not really involved in the adhesion process, but that adhesion molecules can be produced by specific cells in the ectoderm [46]. Although the participation of nematocysts in the adhesion process might be controversial, these studies have shown that several types of nematocysts might be present in the pedal tissue of sea anemones [46,47,48]; therefore, we suggest that the preferential presence of such serine peptidase inhibitors in this region might indicate the participation of such toxins as a defensive mechanism used during competition over the substrate.

In this study, MSI shows that the inhibitors ACPI-II and ACPI-III, with *m*/*z* values of 2633 and 2642, respectively, are primarily present in the mesenteric tissue region. On the other hand, the F3.7.5 fraction peptides, with *m*/*z* values of 4518, 4685, and 4755, and the ACPI-I from the 3.7.6 fraction, with *m*/*z* value of 4461, are predominantly found in the tentacles and pedal disc regions of *A. cascaia*. The differential distribution of toxins in different regions of the sea anemone is expected, as these animals have nematocysts in the tentacles used for predation and defense. Additionally, the presence of nematocytes and gland cells, which are considered toxin reservoirs, have been recorded in the ectoderm and endoderm [1,2]. Mesenterial filaments, which are involved in digestion and defense, have been reported to contain venom components [45,49].

Previous studies have reported the use of MSI to detect the presence of Kunitz peptides in the mesenterial filaments of *A. tenebrosa*, and MALDI-MSI has been employed to elucidate the spatial distribution of κ-actitoxin-Ate1a in *A. tenebrosa*, as well as to localize peptide toxins in the sea anemone *Oulactis muscosa* [6,34,50,51]

Traditionally, the proteome or transcriptome profile of venomous animals has been elucidated using secretions or tissue extracts, which enable the identification of toxins or toxin-like transcripts and reveal the diversity of venoms [30,50,52]. However, such approaches typically result in the loss of spatial distribution of toxins. Combining MSI with histology provides a promising alternative to address this issue. Accessing the localization of toxins in sea anemones, like other cnidarians, can be challenging, given that these animals have venomous capsules distributed throughout their bodies rather than a centralized venom system [50]. These capsules, known as cnidocysts or “thread capsules”, are produced by the Golgi apparatus of specialized cells known as cnidocytes and can contain different toxin compositions depending on the tissue in which they are found [1,45].

It is worth noting that a significant variation in the abundance of toxin-like genes has been reported across the tentacles, mesenterial filaments, and columns of different sea anemone species. For instance, Kunitz protease inhibitors/type II KTx are highly expressed in mesenterial filaments and tentacles of sea anemones, as shown in studies involving *Anemonia sulcata*, *Heteractis crispa*, and *Megalactis griffithsi* species [45]. This finding highlights the importance of investigating the spatial distribution of toxins and toxin-like genes in different regions of sea anemones and other venomous animals. While previous studies have typically used secretions or tissue extracts to identify toxins or toxin-like transcripts, these approaches do not preserve the spatial distribution of the toxins. Combining MSI with histology may provide a promising alternative to overcome this limitation.

In conclusion, this study provides the first evidence of the distribution of serine peptidase inhibitors in the tentacles, pedal disc, and mesenterial filaments of the Brazilian sea anemone *A. cascaia*. The toxins isolated, including ACPI-I, ACPI-II, ACPI-III, and those found in the fraction 3.7.5, showed potent inhibition against trypsin, indicating their potential as interesting molecules for further property characterization. These findings expand our understanding of the ecological roles of venom serine proteases in *A. cascaia* and suggest that these inhibitors may serve multiple functions beyond just defense.

Overall, this work highlights the importance of using MSI as a complementary approach to traditional methods for the study of venomous animals, as it allows for the visualization of the spatial distribution of toxins in different tissues. The results also demonstrate the value of investigating venomous organisms in understudied regions like Brazil, which is home to a diverse range of venomous species. Further research is needed to fully elucidate the functional roles of these toxins and their potential applications in biotechnology and pharmacology.

## 4. Materials and Methods

### 4.1. Animal Collection and Sample Attainment

Adult *A. cascaia* specimens were collected from Cabelo Gordo (23°49′43″ S, 45°25′24″ W) and Cigarras beaches (23°43′55″ S, 45°23′58″ W) at São Sebastião, SP, Brazil, with authorization n° 72666-1 from SISBIO. A total of 10 specimens were collected from the intertidal zone during low tide periods and immediately transported to the Centro de Biologia Marinha (CEBIMar) at Universidade de São Paulo. The specimens were washed with seawater from Cigarras beach and exposed to 0.1% acetic acid for 24 h to induce nematocyst discharge. The resulting venom solution (50 mL) was lyophilized, resuspended in ultrapure water, and stored at −20 °C for further analysis. The venom was then centrifuged at 10,000× *g*/10 min at 5 °C and divided into a precipitate fraction and a soluble fraction (supernatant). The soluble fraction was used for the purification of inhibitors.

### 4.2. Venom Fractionation

The fractionation of venom components was performed by reversed-phase high-performance liquid chromatography (RP-HPLC) using an LC-20A Prominence HPLC system (Shimadzu Co., Kyoto, Japan) with a C18 column (Luna C18(2), 250 × 4.6 mm, 5 μm, 100 Å). Separation was achieved by using a linear gradient (0–100% of B) for 30 min, after 5 min isocratic elution (0% B), at a 1 mL/min constant flow. The employed buffers were A (0.1% acetic acid) and B (0.1% acetic acid in 90% acetonitrile). Eluates were monitored by a Shimadzu detector SPD-M20A PDA; the wavelengths 280 nm—used to detect tryptophan and tyrosine in the sample—and 214 nm, which is absorbed by the peptide bond, were used so that peptides lacking trp and/or tyr were not missed. Fractions were manually collected every 5 min and then lyophilized for later use. Optimized gradients were developed for further separation of active fractions: for F3, a gradient of 5–40% of B in 40 min was used, and for F3.7 and F3.8, a 5–60% gradient of B in 35 min was used.

### 4.3. SDS-PAGE

The electrophoretic profiles of the venom and RP-HPLC fractions were analyzed by SDS-PAGE under reducing conditions, following the protocol established by Laemmli (1970) [53]. Samples were loaded onto polyacrylamide gels with a stacking gel (5%) and resolving gel (12%) and subjected to electrophoresis under 100 V for 1 h. To estimate molecular mass, the ‘Amersham Low Molecular Weight Calibration Kit for SDS Electrophoresis’ (GE Healthcare, Chicago, IL, USA) was used. The gels were stained with Coomassie brilliant blue for visualization.

### 4.4. Enzymatic Assays

Hydrolysis of the fluorogenic peptidyl substrate (0.083 mM; Z-Phe-Arg-MCA) at 30 °C in 100 mM Tris-HCl, pH 8.5, was followed by measuring the fluorescence at λex = 330 nm and λem = 430 nm in a Molecular Devices M2 fluorometer, using 96-well microtiter plates. Trypsin assays were performed in 155 µL of the substrate solution and placed in each well in a thermostatically controlled compartment for 5 min before the enzyme solution was added, and the increase in fluorescence with time was continuously recorded for 30 min. The enzyme concentrations for initial rate determinations were chosen at a level intended to allow less than 5% hydrolysis of the substrate consumption. Qualitative inhibition assays were performed by adding 15 µL of each fraction with trypsin 30 min prior to the addition of substrate solution, and the assay was followed as previously described. Due to the absence of an efficient method for peptide quantification, it was established that 15 µL of each venom fraction (containing approximately 400 µL) would be used for testing the inhibition of trypsin activity. All assays were performed in duplicate. Relative inhibitory activity was determined by calculating the percentage of inhibition in comparison to the positive control. The slope of fluorescence increase per minute was calculated, and linear regression analyses were performed for positive control and assays containing inhibitory fractions. The slopes’ values were compared to estimate the inhibition activity.

### 4.5. Mass Spectrometry Measurements

For Matrix Assisted Laser Desorption Ionization–Time of flight (MALDI-TOF) analysis, 1 µL of the venom/inhibitory fractions were spotted with 1 µL of saturated CHCA (α-cyano-4-hydroxycinnamic acid) in 50% ACN, containing 0.1% Trifluoroacetic acid (TFA) or Sinapinic acid matrix on a MALDI plate and analyzed in an AXIMA series MALDI-TOF/MS (Shimadzu, Co., Kyoto, Japan). Peptide mass spectra data were acquired in positive ion linear mode in the range of *m*/*z* 500–15,000.

For LC-MS, previously lyophilized fractions from *A. cascaia* venom were dissolved into 50 µL of acidified water containing 0.1% TFA. Samples were deposited into the 96-well plate of the SIL-20A autosampler for LC-MS analysis in an ion trap time of flight (IT-TOF) mass spectrometer system (Shimadzu, Co., Kyoto, Japan). A total of 10 µL of sample aliquots were injected and subject to an RP-HPLC (20A Prominence system) separation by a Discovery C18 1.5 (2 × 50 mm) column. Buffer A1 containing (0.1% Formic acid) and Buffer B1 containing (0.1% Formic acid; 90% acetonitrile) were used in a linear gradient from 0 to 100% of B in 6 min, under a constant flow rate of 0.2 mL/min. The interface was kept at 4.5 kV and 200 °C. Detector was kept at 1.95 kV. MS spectra were acquired in positive mode, in the 350–1400 *m*/*z* range. Instrument control, data acquisition, and processing were performed by the LCMS Solution suite (Shimadzu).

In order to determine the number of cysteines in each peptide, samples (4 µL) were diluted in 6 µL of 100 mM Ammonium Bicarbonate solution and reduced by the addition of 10 µL of 250 mM DTT. Samples were heated at 95 °C/20 min. After incubation, samples were alkylated by the addition of 10 µL of 200 mM Iodoacetamide and incubated in the dark for 1 h at room temperature. Reagents were cleaned up by C18 ZipTips, and samples were dissolved in 5 µL of 0.1% TFA mixed 1:1 (*v*:*v*) with saturated CHCA (α-cyano-4-hydroxycinnamic acid) and deposited in the sample. Spectra was acquired in positive ion linear mode in the range of *m*/*z* 1000–10,000.

### 4.6. Proteomic Assays

Previously lyophilized samples were suspended in 20 µL of 50 mM Ammonium Bicarbonate and reduced by adding 2 µL of 100 mM DTT at 60 °C for sind30 min. Then, samples were alkylated by adding 2 µL of 200 mM Iodoacetamide at room temperature for 30 min. Reaction was kept protected from light. Samples were digested by trypsin (1 µg, Trypsin Singles, Proteomics Grade, SIGMA-ALDRICH, St. Louis, MO, USA) overnight at 37 °C. The reaction was stopped with 5 µL of acetic acid.

One microliter of the tryptic peptides was subjected to nano-ESI-LC-MS/MS using a Dionex Ultimate 3000 RSLCnano (Thermo Fisher Scientific, Waltham, MA, USA) coupled with Impact II Mass spectrometer (Bruker Daltonics, Bremen, Germany). Fractions were injected in a nano-trap Acclaim PepMap (Dionex-C18, 100 Å, 75 μm × 2 cm) in 2% solvent A2 (0.1% Formic acid) for 2 min, under a 5 µL·min^−1^ flow rate. Elution was performed by a linear gradient of 5–40% of solvent B2 (0.1% Formic acid in acetonitrile) in 120 min, under 350 nL min^−1^. Mass spectra were acquired in positive mode. MS and MS/MS scans were acquired at 2 Hz, in a *m*/*z* 50–2000 range. CID energy ramped between 7 and 70 eV. Data were processed by Peaks Studio 7.0 (Bioinformatics Solution Inc., Waterloo, ON, Canada), and data were searched against Cnidaria Database from UniProt. Identified peptides were submitted to Blastp searches (NCBI, ‘non-reduntant protein sequences’ and ‘Sea Anemone Taxid 6073’ organism).

### 4.7. MALDI-TOF Mass Spectrometry Imaging (MSI)

A specimen of *A. cascaia* was divided by longitudinal sectioning and left in fixative (5 mL) and ethanol 100% (3 mL) solution for 16 h at 8 °C. The formalin-free fixative used was prepared in-house, as previously described [54]. For dehydration, the tissue was submitted to gradual concentrations of xylene/ethanol (10 to 100%) with intervals of 30 min at each concentration until the tissue was completely immersed in 100% xylene. Subsequently, the tissue was submitted to gradual baths (10 to 100%) of Paraplast^®^ (Oxford Labware, St. Louis, MO, USA) until it was completely embedded. The sample was kept at −20 °C.

MALDI-MSI tissue samples were longitudinally sectioned at 7 µm thickness on a microtome. Sections were placed on ITO (Indium Tin Oxide)-coated glass slides and heated at 58 °C on a heat block for melting paraffin. Sequentially, tissue sections were gently washed with 100% xylene for paraffin removal and images of the tissue were acquired on optical microscope. A Bruker ImagePrep equipment was used for automatic CHCA spraying (105 mg CHCA, 8 mL acetonitrile, 7 mL ultrapure water and 30 µL TFA) over tissue. The slides were placed on Bruker SCiLS™ Lab MTP Slide Adapter II and analyzed in a MALDI-TOF/TOF Ultraflex III mass spectrometer (Bruker, Germany). The MSI analysis was performed under linear positive mode, *m*/*z* range 1000–10,000. Instrument control and data acquisition was performed by FlexControl 3.3. Data parameters were 60 µm spatial resolution, Laser 3-medium, laser power 48%, and 400 shots; matrix ion suppression up to *m*/*z* 980 was used for Geometry determination, and data location was performed by FlexImaging 4.0 (Bruker, Germany), comprising mainly the tentacles, mesenteric tissue and pedal disc of the animal. SciLS™ Lab MS software (Bruker, Bremen, Germany) was used for visualization of two-dimensional (2D) ion-intensity map data. Baseline convolution and root mean square normalization parameters were applied to the spectra acquired. Following MSI analysis, CHCA matrix was removed from tissues and samples were stained with hematoxylin and eosin, according to previously described protocol [55].

## Figures and Tables

**Figure 1 marinedrugs-21-00481-f001:**
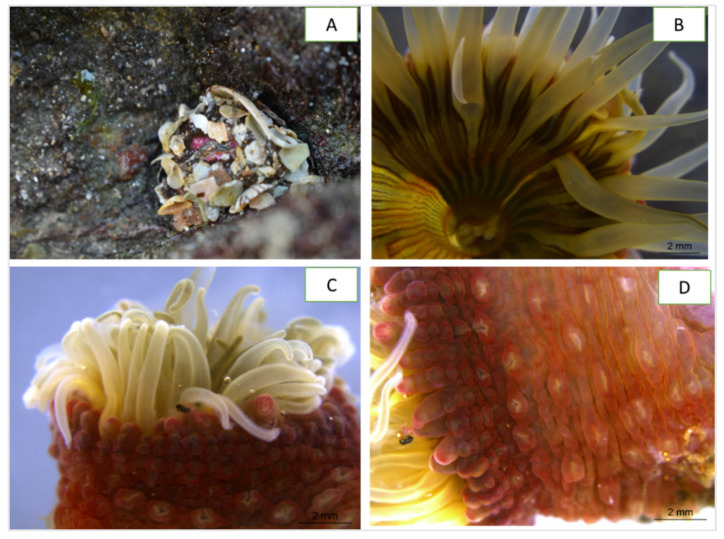
(**A**) *Anthopleura cascaia* sea anemone covered by rocks and shells at the intertidal zone of Cigarras beach, São Sebastião district, São Paulo, Brazil. (**B**) Superior image of oral disc and tentacles from *A. cascaia*. (**C**,**D**) External morphology of a living specimen of *A. cascaia*. Tentacles are visible at superior region of the animal, and adhesive vesicles are distributed throughout the body. (**B**–**D**) images were acquired using a stereo microscope Leica M205 A attached to DMC2900 camera and LAS V4.6 software at Instituto Butantan, São Paulo, Brazil, in 2019.

**Figure 2 marinedrugs-21-00481-f002:**
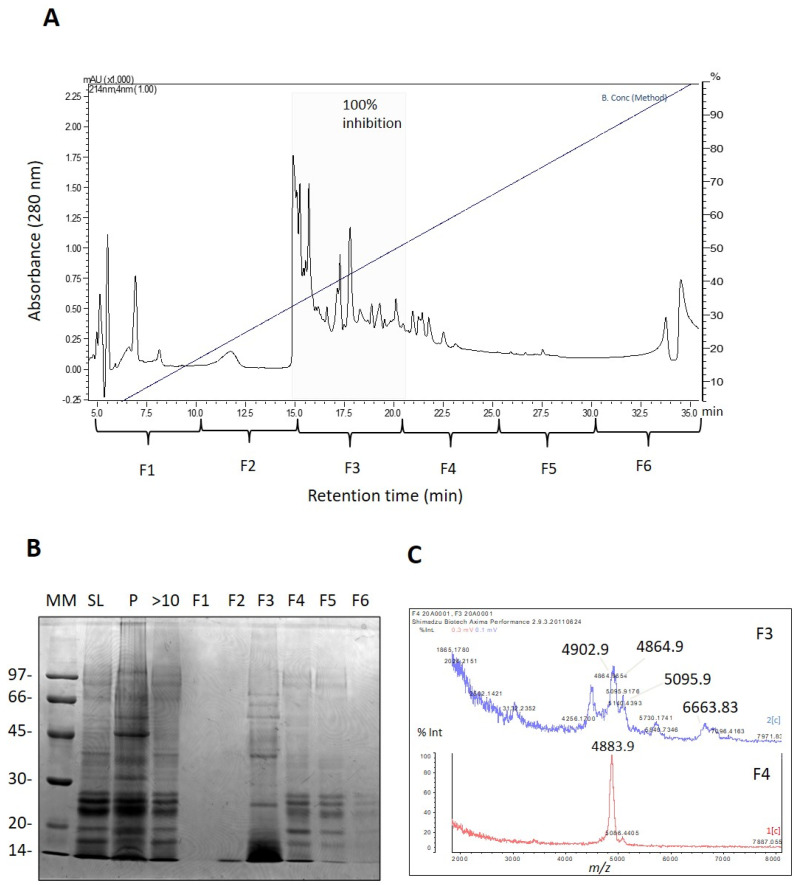
(**A**) Reversed-phase chromatographic separation of *A. cascaia*’s venom. The venom was separated by RP-HPLC using a gradient of 5 to 40% of B (0.1% acetic acid in 90% acetonitrile) in 40 min. Buffer A (0.1% acetic acid) and B (0.1% acetic acid in 90% acetonitrile) were used for separation. Fractions (F1 to F6) were tested regarding trypsin activity, and F3 presented 100% inhibition. (**B**) SDS-PAGE analysis. Samples (20 µL) were analyzed by SDS-PAGE (12%) and stained with Coomassie brilliant blue. The gel presents the pattern of proteins found in the soluble fraction (SL) of the venom, venom precipitate (P), fractions higher than 10 kDa, and HPLC fractions (F1–F6) from *A. cascaia*’s venom. (**C**) MALDI-TOF analysis of venom’s fractions from *A. cascaia* venom. The mass spectra show *m*/*z* ranging from 2000 to 8000, with peaks of *m*/*z* 4864, 4902, 5095, and 6663 corresponding to the most abundant ions in F3 and the peak *m*/*z* 4883 corresponding to the most abundant ion in F4.

**Figure 3 marinedrugs-21-00481-f003:**
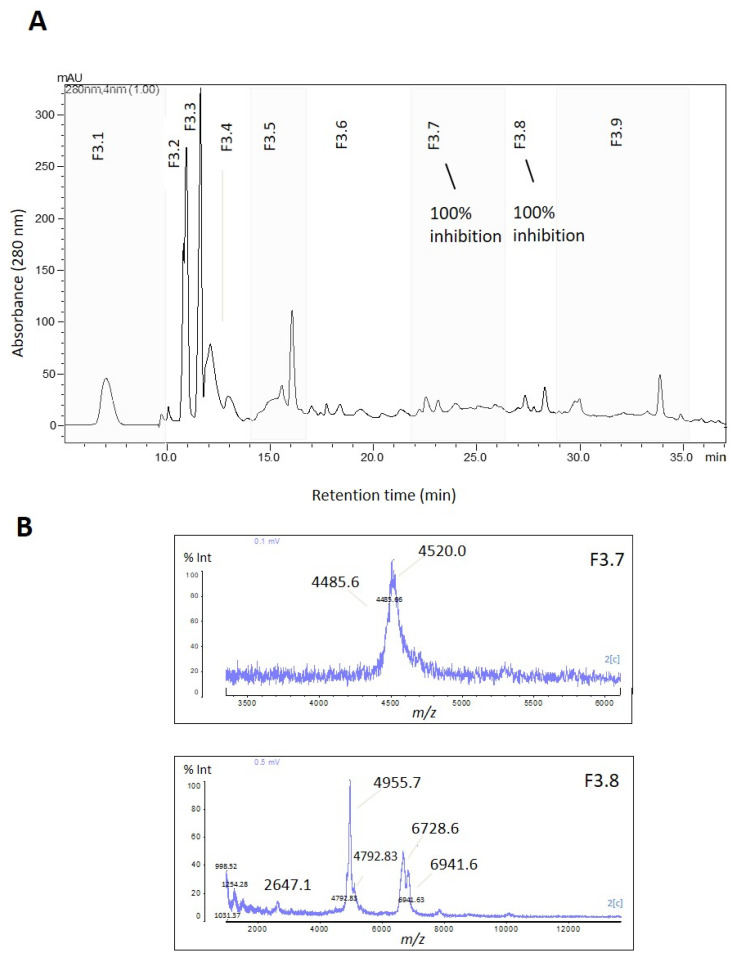
(**A**) Reversed-phase chromatographic separation of Fraction 3 from *A. cascaia*’s venom. The inhibitory fraction was separated by RP-HPLC using a C18 column and a gradient of 5–40% of B in 40 min, leading to the obtention of 9 fractions (3.1 to 3.9). The fractions were tested regarding trypsin inhibition, and F3.7 and F3.8 exhibited 100% enzyme activity inhibition. (**B**) MALDI-TOF analysis of inhibitory fractions (3.7 and 3.8) from *A. cascaia* venom. Fraction 3.7 shows mainly the existence of two components, *m*/*z* 4520 and *m*/*z* 4485, and F3.8 shows *m*/*z* ranging from 2647 to 6941.

**Figure 4 marinedrugs-21-00481-f004:**
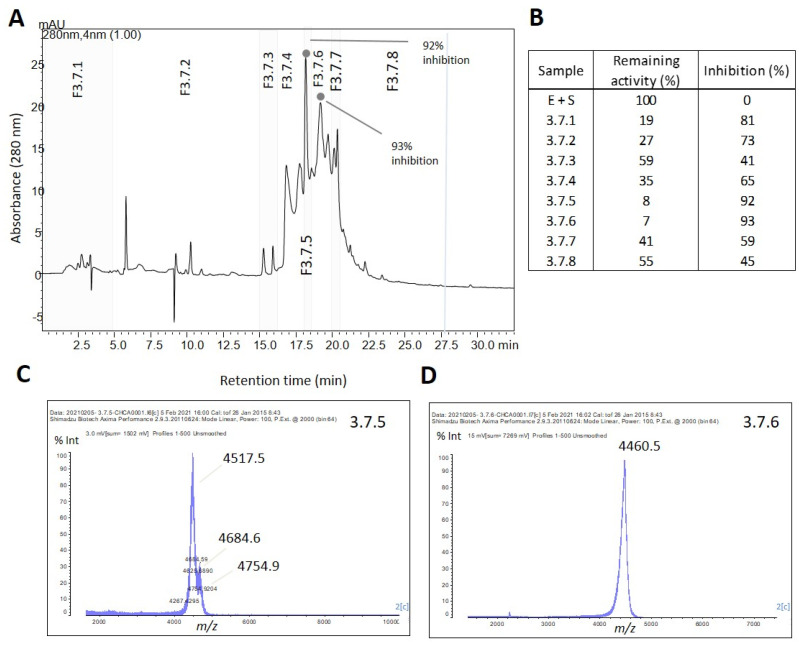
(**A**) Reversed-phase chromatographic separation of Fraction 3.7 from *A. cascaia*’s venom. The fraction was separated by RP-HPLC using a C18 column and a gradient of 5–60% of B in 35 min. The separation led to the obtention of 8 fractions (3.7.1 to 3.7.8) that were subsequently tested regarding trypsin inhibition. F3.7.5 and F3.7.6 presented more than 90% of activity inhibition. (**B**) Trypsin inhibition assay. F3.7 subfractions (5 µL) were incubated for 30 min with trypsin (1:1), and afterward substrate was added to the samples. Substrate consumption in Arbitrary Units of Fluorescence (*Y* axis) was read every 5 min for 35 min (*X* axis). In the test, samples F3.7.1 to F3.7.8 representing venom fractions from *A. cascaia*, Blank (B + S), and positive control (E + S) were analyzed. B = Buffer, S = Substrate, and E = Enzyme. The subfractions F3.7.5 and 3.7.6 exhibited the highest enzyme inhibition (>90%), reflecting lower substrate consumption over time. (**C**,**D**) MALDI-TOF analysis of *A. cascaia* venom fractions. The spectra of F3.7.5 (**C**) by positive mode show the existence of three main components: *m*/*z* 4518, *m*/*z* 4685, and *m*/*z* 4755. Fraction 3.7.6 (**D**) shows the presence of a unique peptide of *m*/*z* 4461.

**Figure 5 marinedrugs-21-00481-f005:**
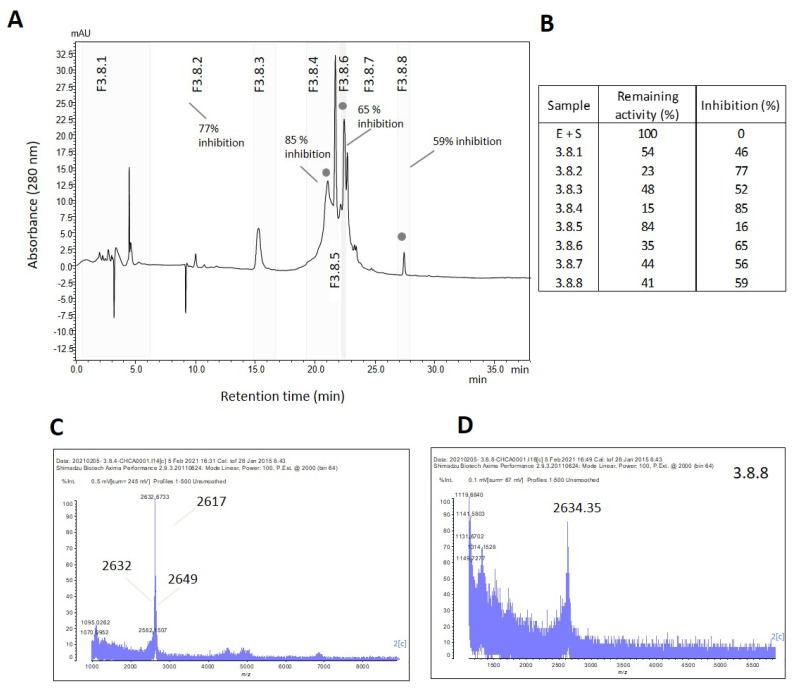
(**A**) Reversed-phase chromatographic separation of Fraction 3.8 from *A. cascaia*’s venom. The fraction was separated by RP-HPLC using a C18 column and a gradient of 5–60% of B in 35 min. The separation led to the obtention of 8 fractions (3.8.1 to 3.8.8). Fractions exhibiting trypsin inhibition are highlighted with a symbol (**°**) above the peaks. (**B**) Trypsin inhibition assay. F3.8 subfractions (5 µL) were incubated for 30 min with trypsin (1:1), and substrate was added to the samples afterward. Substrate consumption in Arbitrary Units of Fluorescence (*Y* axis) was read every 5 min for 35 min (*X* axis). In the test, samples F3.8.1 to F3.8.8 representing venom fractions from *A. cascaia*, Blank (B + S), and positive control (E + S) were analyzed. B = Buffer, S = Substrate, and E = Enzyme. F3.8.4, F3.8.6, and 3.8.8 exhibited the highest enzyme inhibition, reflecting lower substrate consumption over time. (**C**,**D**) MALDI-TOF analysis of *A. cascaia* venom fractions. (**C**) The spectra of F3.8.4 by positive linear mode shows the existence of *m*/*z* 2617, 2633 and 2649. (**D**) Fraction 3.8.8 shows the presence of a main mass of *m*/*z* 2634.3.

**Figure 6 marinedrugs-21-00481-f006:**
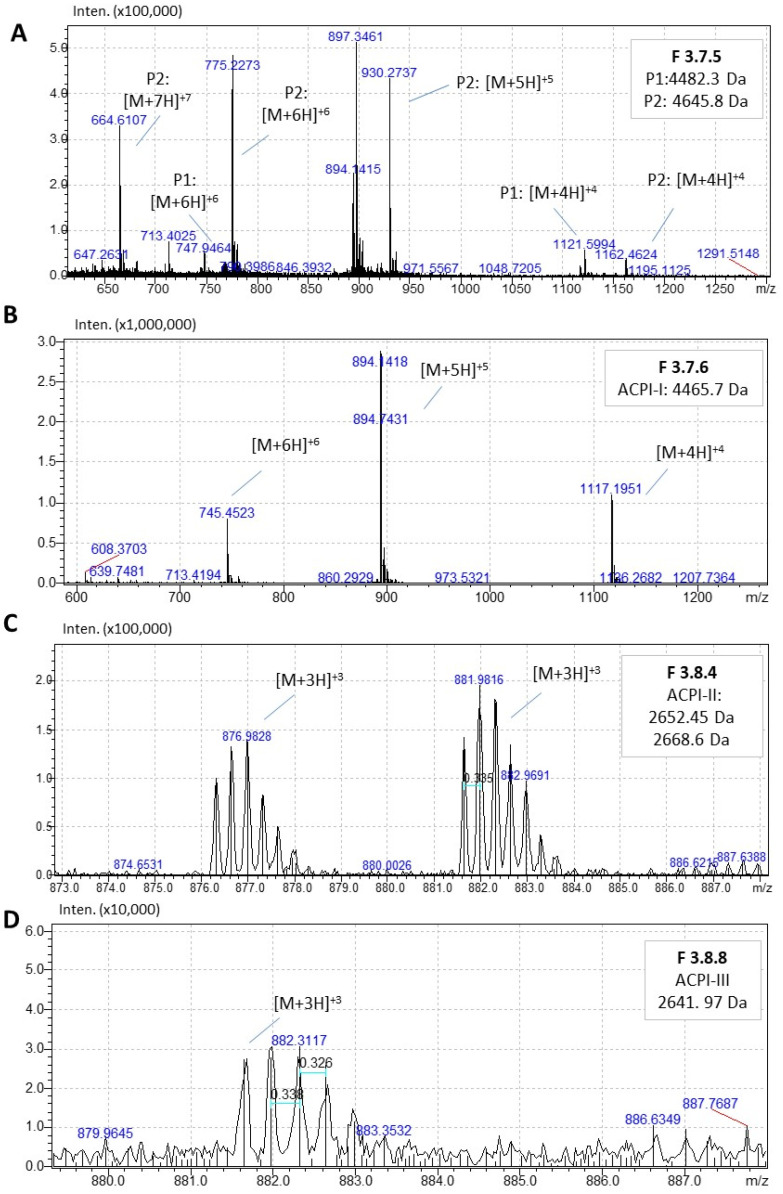
Mass profile of serine peptidase inhibitors isolated from venom’s fractions by LC-MS using ESI-IT-TOF. (**A**) Shows the *m*/*z* corresponding to two peptide inhibitors (P1 and P2) found in F3.7.5. (**B**) Shows the *m*/*z* of the purified inhibitor ACPI-I from F3.7.6. (**C**) Shows the detection of two *m*/*z* with 16 Da difference (oxidation) corresponding to ACPI-II, a peptide isolated from F3.8.4. (**D**) Shows the detection of *m*/*z* corresponding to ACPI-III isolated from F3.8.8.

**Figure 7 marinedrugs-21-00481-f007:**
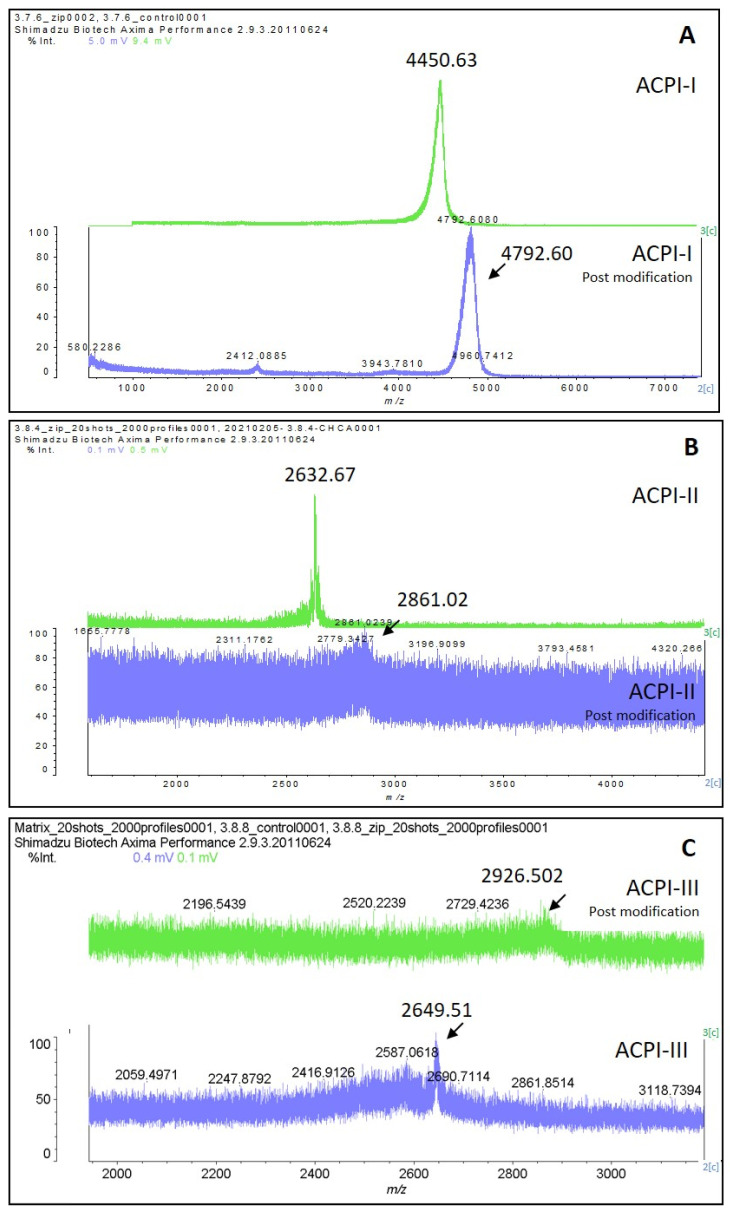
Identification of cysteines in ACPI-I (**A**), ACPI-II (**B**), and ACPI-III (**C**) peptides by MALDI-TOF. Isolated peptides were reduced using DTT and alkylated by Iodoacetamide, presenting an increase of +57 Da per cysteine found in the sample. Each spectrum shows the intact peptide *m*/*z* (green color) and the respective *m*/*z* post modification of cysteines. (**A**) Shows a mass difference of 342 in ACPI-I post modification (blue spectrum), signalizing the existence of 6 cysteines in this peptide. (**B**) Shows a mass increase of *m*/*z* 228 in ACPI-II (blue spectrum), revealing 4 cysteines in ACPI-II. (**C**) Shows the intact mass of ACPI-III *m*/*z* (blue color) and the respective *m*/*z* post modification of cysteines (green color). This figure shows a mass increase of 277 in ACPI-III, presenting at least 4 cysteines in this peptide.

**Figure 8 marinedrugs-21-00481-f008:**
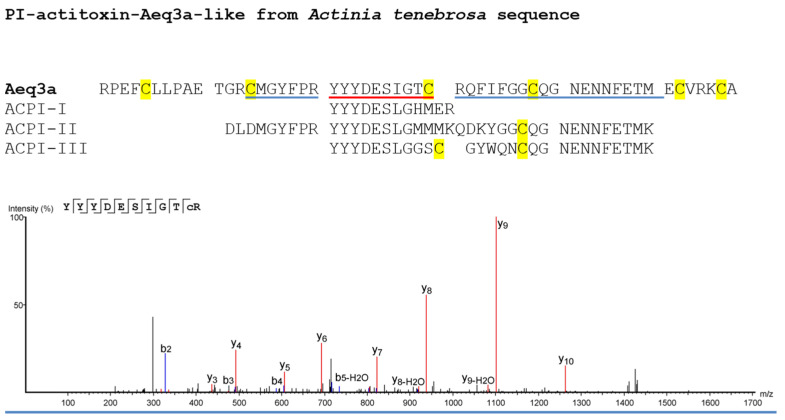
Coverage of Blastp hit PI-actitoxin-Aeq3a-like and De novo peptides identified in ACPI-I, ACPI-II, and ACPI-III. The figure shows the amino acid (aa) sequence of PI-actitoxin-Aeq3a-like from *A. tenebrosa*. Peptides identified by LC-MS/MS are highlighted in blue and red. The *m*/*z* corresponding to the sequence ‘YYYDESIGTC’ (highlighted in red) was identified in all three purified peptides (ACPI-I, ACPI-II, and ACPI-III) as seen in Appendix A. The respective mass spectrum from this aa sequence is shown below the sequences, exhibiting y and b ions distribution. De Novo analysis based on the mass spectra acquired for each purified peptide (ACPI-I, ACPI-II, and ACPI-III) suggests new amino acid sequences that exhibit similarity to actitoxin-Aeq3a-like scaffold. The suggested de novo sequences are shown below Aeq3a-like toxin. Cysteine residues that characterize Kunitz domain are highlighted in yellow. Average Local Confidence percentages (ALC %) from De novo peptides are shown in Appendix A, and Mass spectra are shown in Appendix A.

**Figure 9 marinedrugs-21-00481-f009:**
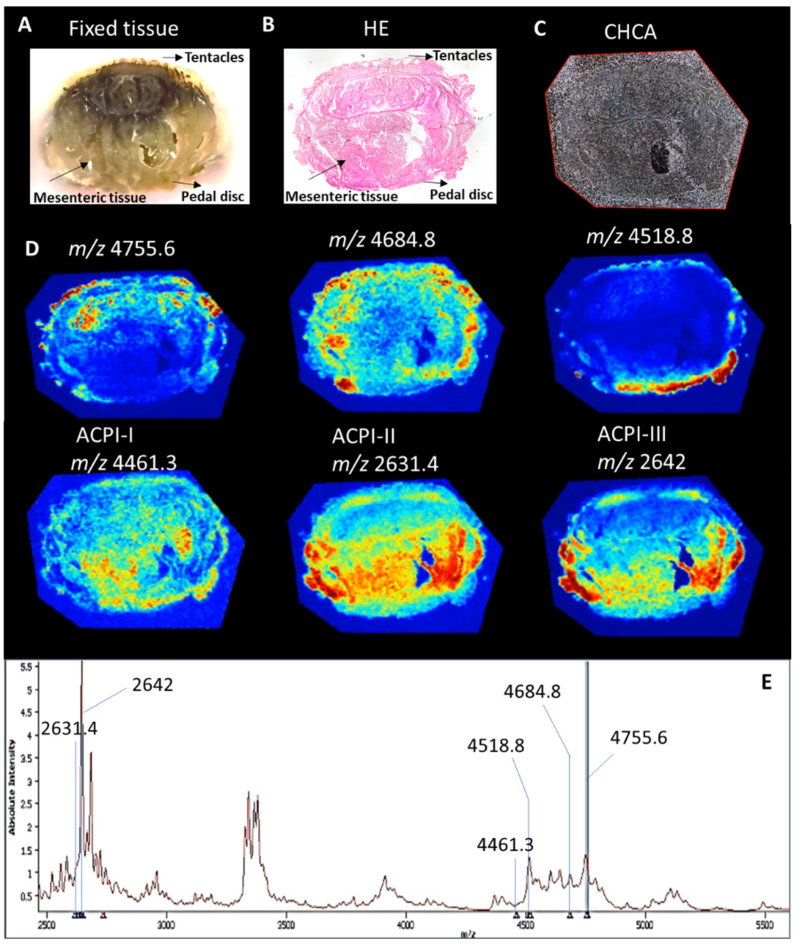
(**A**) MALDI-MS imaging–spatial distribution of serine peptidase inhibitors on *A. cascaia*’s tissue. (**A**) Image of paraffinized tissue from *A. cascaia*. (**B**) Histological image of the sea anemone longitudinal section used for MSI, stained with hematoxylin and eosin. (**C**) Tissue section sprayed with matrix CHCA. (**D**) Heat map showing the distribution of serine peptidase inhibitors from F 3.7.5 (*m*/*z* 4755.6, *m*/*z* 4684.8, and *m*/*z* 4518.8), ACPI-I (*m*/*z* 4461.3), ACPI-II (*m*/*z* 2631.4), and ACPI-III (*m*/*z* 2642) ions across *A. cascaia*’s tissue. The ions localization is represented by hotspots. (**E**) Positive mode average spectra acquired in the longitudinal section of the animal. *Y* axis corresponds to the signal intensity. *X* axis shows the corresponding *m*/*z*.

**Table 1 marinedrugs-21-00481-t001:** LC-MS/MS analysis from ACPI-I, ACPI-II, and ACPI-III. The table presents the BLASTp hit (Aeq3a) identified for all three peptides isolated from *A. cascaia* venom. Peptides isolated from *A. cascaia* were digested with trypsin and analyzed by LC-MS/MS. Mass spectra acquired were searched against Cnidaria database from UniProt. The coverage of Blastp hit, −10 lgP, and supporting peptides identified followed by the respective −10 lgP are shown according to the molecule analyzed. Supporting peptides may present Cysteine (C) residues with +57.02 mass shift due to alkylation with Iodoacetamide. Methionine (M) residues from peptides also present a mass increase of 15.99 due to oxidation.

Isolated Peptide	Blastp Hit/Species/Accession	−10 lgP	* Cov (%)	Supporting Peptides/−10 lgP
ACPI-I	PI-actitoxin-Aeq3a-like/ *Actinia tenebrosa*/A0A6P8HBL9_ACTTE	84.75	37	R.YYYDESIGTC(+57.02)R.Q/75.41 R.QFIFGGC(+57.02)QGNENNFETM(+15.99)K.E/18.67
ACPI-II	PI-actitoxin-Aeq3a-like/ *Actinia tenebrosa*/A0A6P8HBL9_ACTTE	107.02	46	R.YYYDESIGTC(+57.02)R.Q/78.33; R.C(+57.02)M(+15.99)GYFPR.Y/45.83; R.QFIFGGC(+57.02)QGNENNFETMK.E/17.32
ACPI-III	PI-actitoxin-Aeq3a-like/ *Actinia tenebrosa*/A0A6P8HBL9_ACTTE	80.81	14	R.YYYDESIGTC(+57.02)R.Q/80.81

* Coverage.

## Data Availability

The raw data that support the findings of this study are available from the corresponding author, D.L.d.S., upon reasonable request.

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
