# Peer review of "Spatial Distribution and Biochemical Characterization of Serine Peptidase Inhibitors in the Venom of the Brazilian Sea Anemone Anthopleura cascaia Using Mass Spectrometry Imaging"

_marinedrugs, 2023, doi:10.3390/md21090481_

Round 1

Reviewer 1 Report

In the literature, numerous reports on protease inhibitors of marine origin exist. These molecules have great potential as molecular tools in various fields of biomedicine. da Silva et al. present a manuscript where protease inhibitors from a sea anemone were identified, contributing to the knowledge of protease inhibitors. However, the following comments should be addressed before the publication of the manuscript.

After placing the organisms in 0.1% acetic acid solution, how did you preserve them? How many organisms did you collect to perform this research?

The units separate them from the numerical value, in line 465, it says mL.min; it should say mL min or mL/min

Write the meaning of IAA

How did you identify  the organisms?

Figure 1. Indicate the data software, city, and year.

In the supplementary material, check that the axes  contain names and units.

It is risky to propose that due to the presence and distribution of protease inhibitors in the pedal disc of the anemone, these molecules may participate in the adhesion capacity of the organism. In this sense, the authors requested to support this part with experimental data from other techniques or published literature. Likewise, the possible role of protease inhibitors must be discussed.

Various articles have reported the ecological importance and their application of protease inhibitors. However, to have more evidence of the potentiality of the protease inhibitors identified in this manuscript, showing the value of the inhibition constant would reinforce the possible application of such inhibitors.

Check the grammar of the manuscript.

Author Response

Thank you for the opportunity of submitting a revised version of our manuscript. Below, you can find all the points that were raised during the review process, and our replies, in the same order they were presented to us.

In the literature, numerous reports on protease inhibitors of marine origin exist. These molecules have great potential as molecular tools in various fields of biomedicine. da Silva et al. present a manuscript where protease inhibitors from a sea anemone were identified, contributing to the knowledge of protease inhibitors. However, the following comments should be addressed before the publication of the manuscript.

After placing the organisms in 0.1% acetic acid solution, how did you preserve them?

Organisms were placed in 0.1% acetic solution for 24 h. The animals did not survive.

How many organisms did you collect to perform this research?

We have collected 10 specimens for venom extraction, during the course of this study. Another 2 specimens that were collected in 2020 and properly preserved, were employed for tissue preparation for Mass Spectrometry Imaging.

The units separate them from the numerical value, in line 465, it says mL.min; it should say mL min or mL/min

Reviewed.

Write the meaning of IAA

Reviewed. IAA means Iodoacetamide. IAA was replaced for Iodoacetamide in the manuscript.

How did you identify the organisms?

Organisms were identified in the field due to the characteristic presence of small particles of shells attached to their retracted column (therefore the name: cascaia relates to ‘cascalho’, which means ‘gravel’ in Portuguese, as described by Correa, 1973). These sea anemones have characteristic adhesive vesicles throughout their bodies, which allows them to use these small shells and rocks particles for mixing up with the surrounding environment. After collecting these animals, the specimens were taken to the lab and were identified by the zoologist, expert in Cnidaria – Dr. Andre Carrara Morandini, Zoology professor at the Universidade de Sao Paulo - Brazil.

Figure 1. Indicate the data software, city, and year.

Legend of Fig 1 was updated to: ‘B, C and D images were acquired using a stereo microscope Leica M205 A attached to DMC2900 camera and LAS V4.6 software at Instituto Butantan, São Paulo – Brazil in 2019.’

In the supplementary material, check that the axes contain names and units.

Reviewed.

It is risky to propose that due to the presence and distribution of protease inhibitors in the pedal disc of the anemone, these molecules may participate in the adhesion capacity of the organism. In this sense, the authors requested to support this part with experimental data from other techniques or published literature. Likewise, the possible role of protease inhibitors must be discussed.

We are simply describing a causal correlation. The pedal disc is used for adhesion. There are inhibitors present in the pedal disc. Therefore, they might be involved in the process. It is our understanding that the final section of the discussion is the proper place to speculate possible physiological roles for the identified molecules, based on the many results we have presented. Otherwise, the manuscript becomes too descriptive.

Various articles have reported the ecological importance and their application of protease inhibitors. However, to have more evidence of the potentiality of the protease inhibitors identified in this manuscript, showing the value of the inhibition constant would reinforce the possible application of such inhibitors.

Unfortunately, we were not able to gather enough peptides to proper quantify them. Consequently, no Ki could be calculated. We limited ourselves to present qualitative data regarding the kinetic assays.

Check the grammar of the manuscript.

Reviewed.

Reviewer 2 Report

The paper by da Silva and collaborators deals with the physiological distribution of a series of toxins, specifically serin protease inhibitors, within the body of Anthopleura cascaia using mass spectroscopy imaging. After the obtention of the venom, this was fractionated by RP-HPLC, and then its activity was followed by trypsin inhibition. After the identification of the toxins, the abundance of these in the sea anemone body was tracked using mass spectroscopy imaging.

The overall outline of the paper is fine. However, some experimental decisions and procedures are unclear. I think the authors should address and/or comment some key points and questions listed below. I recommend major revisions.

Major concerns:

1.     Trypsin is the serine peptidase chosen as model to test the inhibitory effects of the obtained toxins. However, this enzyme is found in the digestive systems of vertebrates, where it is unlikely to be encountered in the wild by these inhibitors. The authors should further justify the election of this model enzyme.

2.     It is claimed that the isolated inhibitors are capable of binding trypsin, reducing its activity. The inhibition is shown using a fluorescence assay. However, the concentration of the inhibitors is unclear. The authors should try to quantify the amount of inhibitor used in each assay. Otherwise, the comparison “The peptides exhibited inhibition ranging from 59% to 93%” (line 28) is of little meaning, as it might be result of just a difference in concentration. This is of great importance regarding the conclusion outlined in lines 399-401.

3.     How was the percentage of inhibition calculated? This should be indicated.

4.     The criteria used for the separation of the secondary fractions, as for F3.1 to F3.9, is unclear. For example, F3.1 contains one peak. However, F3.4 clearly contains two peaks, as F3.8. But the volumes, presumably, are also different. The criteria should be clearly stated.

Other comments:

1.     Since the authors were able to purify (at least some of) these toxins, it would be nice to show further evidence for the binding of these molecules to trypsin under controlled conditions of concentration. Isothermal titration calorimetry would be a nice technique for this, if it available to the authors and it is feasible with the toxin amounts purified. Alternatively, a dose dependent inhibition assay would also be helpful to further understand the results.

2.     Figure 3B: the two components indicated are greatly overlapping. Furthermore, the complexity of the peak is then resolved and shown in Fig. 4. Hence, there is no need to specify the two components of a complex peak that later is shown to be a composite of more than two species. 

3.     Line 41: extra pair of brackets (“[[1,2]]”).

4.     World Register of Marine Species is abbreviated at least once as WoRMS (line 74), and at least once as WORMS (line 85). The abbreviation in line 74 is the one featured on the website. Please correct.

5.     Figure 2A: Axis labels are hard to read. 

6.     Figure 2B: in the caption it is indicated (line 158) that P is “venom precipitate”. I cannot find P in the gel. Presumably, the authors meant “SD” instead. Labels are again small.

7.     Figure 2C: Is the x-scale the same for F3 and F4? Besides, all labels are too small and very hard to read. Furthermore, the m/z values of many peaks are indicated with labels. These labels are too small, not readable, and not used later. Highlighting just the ones mentioned in the caption would probably be enough and clearer.

8.     Figure 3A: the rectangle around “F3.2” could be set to transparent.

9.     Figure 4: labels cannot be read. Please modify the plots, specially 4C and 4D.

10.  Line 300: ALC (%) is not defined.

11.  Data in Figure 4B and Figure 5B should be presented as a Table (if possible).

12.  Line 327-328: entry numbers are reported. These are, I guess, UniProt entry numbers, which should be indicated.

13.  Line 455: please define the composition or origin of “artificial sea water”.

14.  Line 467: the basis for selecting the 214 and 280 nm wavelengths should be outlined.

15.  Line 496: replace “0,1%” with “0.1%” (replace comma).

16.  Lines 496 and 501: TFA abbreviation seems to have been used for two different things? Please correct.

17.  Line 514: I might have missed it, but I think IAA is not defined.

18.  Caption of Figure 9, line 308: “E-“ should be “(E)”.

19.  Graphs in Fig. S2 could be redrawn to be clearer.

20.  Reference 50 is incomplete.

If possible, I would modify the position of the Materials and Methods section. In my opinion, it would be clearer if it was placed before the results section, but it can also be placed after the conclusions. As it is now, between the discussion and the conclusions, it breaks the rhythm of the last part of the paper.

I would also replot all MS traces in figures 2-5 and 7. As mentioned before, there are far too many labels. The authors could also use higher magnification in the m/z axis if they want to show more clearly some of the complex peaks they find, maybe using an insert to show the overall trace and the context of the highlighted peaks.

Author Response

Thank you for the opportunity of submitting a revised version of our manuscript. Below, you can find all the points that were raised during the review process, and our replies, in the same order they were presented to us.

  1. Trypsin is the serine peptidase chosen as model to test the inhibitory effects of the obtained toxins. However, this enzyme is found in the digestive systems of vertebrates, where it is unlikely to be encountered in the wild by these inhibitors. The authors should further justify the election of this model enzyme.

In spite I work currently as a toxinologist, I have a strong background in enzymology, I can assure the referee that trypsin is “the” model for any serine peptidase (for example, Structure and specific binding of trypsin: comparison of inhibited derivatives and a model for substrate binding. J Mol Biol (25-2-1974) 83, 209-230; The 2.2-_ resolution x-ray crystal structure of the complex of trypsin inhibited by 4-chloro-3-ethoxy-7-guanidinoisocoumarin: a proposed model of the thrombin-inhibitor complex J Am Chem Soc (1990) 112, 7783-7789); Inhibition of trypsin and thrombin by amino(4-amidinophenyl)methanephosphonate diphenyl ester derivatives: X-ray structures and molecular models. Biochemistry (12-3-1996) 35, 3147-3155). There are many reasons for that being, but mainly it is because trypsin does not present any active site sterical hindering, therefore making it the ‘minimal serine peptidase’.

  1. It is claimed that the isolated inhibitors are capable of binding trypsin, reducing its activity. The inhibition is shown using a fluorescence assay. However, the concentration of the inhibitors is unclear. The authors should try to quantify the amount of inhibitor used in each assay. Otherwise, the comparison “The peptides exhibited inhibition ranging from 59% to 93%” (line 28) is of little meaning, as it might be result of just a difference in concentration. This is of great importance regarding the conclusion outlined in lines 399-401.

We agree with the referee that determining the inhibitor concentration would be very important. R1 also commented on that. Unfortunately, we were not able to gather enough peptides to proper quantify them. Consequently, no Ki could be calculated. We limited ourselves to present qualitative data regarding the kinetic assays. Therefore, the assays were designed as screening tests of chromatographic fractions aiming to identify the best candidates to be further characterized by mass spectrometry.

  1. How was the percentage of inhibition calculated? This should be indicated.

Relative inhibition, i.e., decreased in the hydrolysis rate (velocity) when compared to the control (same enzyme concentration and same substrate concentration, without inhibitor).

  1. The criteria used for the separation of the secondary fractions, as for F3.1 to F3.9, is unclear. For example, F3.1 contains one peak. However, F3.4 clearly contains two peaks, as F3.8. But the volumes, presumably, are also different. The criteria should be clearly stated.

Peaks were collected according chromatographic distribution and the most possible base-line resolution. Additionally,, intense peaks were – preferably -  collected individually. For peaks that were not well-resolved (very wide or presenting low intensity), fractions were collected using every 2 to 5 min.

Other comments:

1.Since the authors were able to purify (at least some of) these toxins, it would be nice to show further evidence for the binding of these molecules to trypsin under controlled conditions of concentration. Isothermal titration calorimetry would be a nice technique for this, if it available to the authors and it is feasible with the toxin amounts purified. Alternatively, a dose dependent inhibition assay would also be helpful to further understand the results.

Once again, I should call the referee’s attention to the enzymology jargons. The usual way to evaluate the binding of a given molecule to the active site of a given enzyme is by using the Michaelis-Menten (and derived) equations, initially published in “Michaelis L, Menten ML. Die Kinetik der Invertinwirkung. Biochemische Zeitschrift. 1913;49:333–369”. When such interaction is productive, i.e., there is cleavage, the overall affinity is given by the Michaelis-Menten constant (Km). If there is no product formation, or if the rate of cleavage is very low, then an overall inhibition constant may be calculated. It is termed Ki. Unfortunately, the determination of such constants depend on the knowledge of the molecule concentration in the assay.

  1. Figure 3B: the two components indicated are greatly overlapping. Furthermore, the complexity of the peak is then resolved and shown in Fig. 4. Hence, there is no need to specify the two components of a complex peak that later is shown to be a composite of more than two species. 

Reviewed.

  1. Line 41: extra pair of brackets (“[[1,2]]”).

Reviewed.

  1. World Register of Marine Species is abbreviated at least once as WoRMS (line 74), and at least once as WORMS (line 85). The abbreviation in line 74 is the one featured on the website. Please correct.

Reviewed.

  1. Figure 2A: Axis labels are hard to read. 

Reviewed.

  1. Figure 2B: in the caption it is indicated (line 158) that P is “venom precipitate”. I cannot find P in the gel. Presumably, the authors meant “SD” instead. Labels are again small.

Reviewed.

  1. Figure 2C: Is the x-scale the same for F3 and F4? Besides, all labels are too small and very hard to read. Furthermore, the m/zvalues of many peaks are indicated with labels. These labels are too small, not readable, and not used later. Highlighting just the ones mentioned in the caption would probably be enough and clearer.

Yes. It is the same scale. Labels were updated. Main m/z were highlighted as suggested.

  1. Figure 3A: the rectangle around “F3.2” could be set to transparent.

The transparency of rectangles was increased.

  1. Figure 4: labels cannot be read. Please modify the plots, specially 4C and 4D.

Reviewed

  1. Line 300: ALC (%) is not defined.

Reviewed. ALC stands for Average Local Confidence.

  1. Data in Figure 4B and Figure 5B should be presented as a Table (if possible).

Fig 4B and 5B are presented as tables.

  1. Line 327-328: entry numbers are reported. These are, I guess, UniProt entry numbers, which should be indicated.

Reviewed.

  1. Line 455: please define the composition or origin of “artificial sea water”. Animals were washed in sea water.

This sentence is now corrected.

  1. Line 467: the basis for selecting the 214 and 280 nm wavelengths should be outlined.

Biochemistry jargon: 214 nm is the wavelength that the carbon sp3 hybridization absorbs the most light. Such hybridization is present in the peptide bond; therefore, such wavelength is normally used to monitor peptides elution throughout a chromatographic separation. 280 nm, on the other hand, is typical for aromatic rings. In a biochemical context, aromatic rings may be present in the side chain of aromatic amino acids, namely, tryptophan, phenylalanine, tyrosine, and histidine. Moreover, since 280 nm is further from the low UV, there is less influence of dissolved gases in the mobile phase, making the baseline and, consequently, the chromatogram look smoother. However, not all peptides contain such aromatic amino acids, so dual detection is performed in order to maximize the separation.

  1. Line 496: replace “0,1%” with “0.1%” (replace comma).

Reviewed.

  1. Lines 496 and 501: TFA abbreviation seems to have been used for two different things? Please correct.

Reviewed.

  1. Line 514: I might have missed it, but I think IAA is not defined.

Reviewed. IAA stands for Iodoacetamide. IAA was replaced for Iodoacetamide throughout the manuscript.

  1. Caption of Figure 9, line 308: “E-“ should be “(E)”.

Reviewed

  1. Graphs in Fig. S2 could be redrawn to be clearer.

 Graphs have been updated with different colors for each sample, for better visualization.

  1. Reference 50 is incomplete.

Reviewed

If possible, I would modify the position of the Materials and Methods section. In my opinion, it would be clearer if it was placed before the results section, but it can also be placed after the conclusions. As it is now, between the discussion and the conclusions, it breaks the rhythm of the last part of the paper.

It is the Journal’s format. I don’t think it is possible to rearrange the sections.

 I would also replot all MS traces in figures 2-5 and 7. As mentioned before, there are far too many labels. The authors could also use higher magnification in the m/z axis if they want to show more clearly some of the complex peaks they find, maybe using an insert to show the overall trace and the context of the highlighted peaks.

Reviewed. Font size was increased for better visualization.

Round 2

Reviewer 2 Report

Thank you for your replies. They have properly addressed all my concerns.

Still, I will suggest the following minor changes, which essentially imply incorporating some of the info in your replies into the manuscript. The couple of things I would prioritise are:

- As agreed, determining of inhibitor the concentration would be very important, but unfortunately it was not possible. I am of the opinion that this should be explicitly stated in the manuscript.

- As the authors explain, 280 nm wavelength is used to detect aromatic residues. I should point out, however, that only tryptophan and tyrosine, which have molar extinction coefficients at 280 nm of ~5500 and ~1450 (in M^-1 cm^-1 units) absorb at 280, whereas phenylalanine and histidine do not. In any case, I just think it would be beneficial for new readers if the manuscript included a sentence in the line of "280 was used to detect trp and tyr, and 214, which is absorbed by the peptide bond, was used so that peptides lacking trp and/or tyr were not missed".

Author Response

Thank you for your comments.

Lines 554-558  reads: "Qualitative inhibition assays were performed by adding 15 µL of each fraction with trypsin during 30 min prior the addition of substrate solution, and the assay was followed as previously described. Due to the absence of an efficient method for peptide quantification, it was stablished that 15 µL of each venom fraction (containing approximately 400 µL) would be used for testing the inhibition of trypsin activity.

lines  531-534 reads:
Eluates were monitored by a Shimadzu detector SPD-M20A PDA, selecting the wave-length 280nm – used to detect tryptophan and tyrosine in the sample – and 214 nm, which is absorbed by the peptide bond, was used so that peptides lacking trp and/or tyr were not missed.